# Safety and Cross-Neutralizing Immunity Against SARS-CoV-2 Omicron Sub-Variant After a Booster Dose with SOBERANA^®^ Plus in Children and Adolescents

**DOI:** 10.3390/vaccines13121198

**Published:** 2025-11-27

**Authors:** Dagmar García-Rivera, Meiby Rodríguez-González, Beatriz Paredes-Moreno, Rinaldo Puga-Gomez, Yariset Ricardo-Delgado, Carmen Valenzuela Silva, Sonsire Fernández-Castillo, Rocmira Pérez-Nicado, Laura Rodríguez-Noda, Darielys Santana-Mederos, Yanet Climent-Ruiz, Enrique Noa-Romero, Otto Cruz-Sui, Belinda Sánchez-Ramírez, Tays Hernández-García, Ariel Palenzuela-Diaz, Yury Valdés-Balbín, Vicente G. Vérez-Bencomo

**Affiliations:** 1Finlay Vaccine Institute, 200 and 21 Street, Havana 11600, Cuba; mcrodriguez@finlay.edu.cu (M.R.-G.); bparedes@finlay.edu.cu (B.P.-M.); carmenvalenzuelasilva@gmail.com (C.V.S.); rpnicado@finlay.edu.cu (R.P.-N.); lmrodriguez@finlay.edu.cu (L.R.-N.); dsantana@finlay.edu.cu (D.S.-M.); ycliment@finlay.edu.cu (Y.C.-R.); yvbalbin@finlay.edu.cu (Y.V.-B.); vicente.verez@finlay.edu.cu (V.G.V.-B.); 2Pediatric Hospital “Juan Manuel Márquez”, Havana 11500, Cuba; puga@infomed.sld.cu (R.P.-G.); yariricardo@infomed.sld.cu (Y.R.-D.); 3National Civil Defense Research Laboratory, San José de las Lajas 32700, Cuba; dc_virologia@unicom.co.cu (E.N.-R.); dc_cientiprod@unicom.co.cu (O.C.-S.); 4Center for Molecular Immunology, 15th Ave. and 216 Street, Havana 11600, Cuba; belinda@cim.sld.cu (B.S.-R.); tays@cim.sld.cu (T.H.-G.); 5Center of Immunoassays, 134 and 25 Street, Havana 11600, Cuba; ariel.palenzuela@cie.cu

**Keywords:** COVID-19 vaccine, booster dose, Omicron variant, SOBERANA, cross-neutralizing immunity

## Abstract

**Background:** With the emergence of SARS-CoV-2 Omicron sub-variants exhibiting increased transmissibility and immune escape, booster immunization is recommended. Ideally, vaccination across all age groups, including children and adolescents, is critical to control viral spread and reduce variant emergence. The heterologous scheme consisting of two doses of SOBERANA^®^ 02 followed by a third dose of SOBERANA^®^ Plus, which are recombinant protein subunit vaccines constructed from the ancestral RBD, has proven safety, immunogenicity, and effectiveness in pediatric populations as primary series. This study evaluated the safety and immunogenicity of a SOBERANA^®^ Plus booster dose administered six months after primary vaccination in individuals aged 3–18 years. **Methods:** In this follow-up analysis of a phase I/II trial, 244 participants received the booster. Safety was monitored via active surveillance at 1 h, 24 h, and over 28 days post-vaccination. Humoral responses were assessed 28 days post-booster. Antibody responses to the SARS-CoV-2 nucleocapsid (N) protein were assessed in all collected serum samples. **Results:** Adverse events occurred in 18% of participants, predominantly local (85.2%) versus systemic (14.8%); no serious or severe adverse events were reported. All humoral response parameters increased significantly post-booster, including neutralizing antibodies against D614G (24.7-fold increase) and Omicron BA.1 (55.9-fold increase), with similar responses in N-negative and N-positive individuals. Importantly, cross-neutralizing activity against recent Omicron sub-variants (XBB.1.5 and EG.5.1) was also detected. **Conclusions:** A SOBERANA^®^ Plus booster is safe and significantly enhances cross-neutralizing immunity against evolving Omicron sub-variants in children and adolescents. These results highlight the potential of first-generation RBD-based vaccines to maintain broad immunity despite viral evolution.

## 1. Introduction

The ongoing evolution of SARS-CoV-2 has given rise to Omicron subvariants with increased transmissibility and immune escape have spread worldwide [1]. Boosting the immune response acquired through vaccination has been recommended, particularly for the elderly and individuals at higher risk of severe COVID-19 [2]. Although the disease is often asymptomatic or mild in children, they can still transmit the virus to susceptible individuals [3]. Ideally, vaccination and booster doses are needed in all age groups to effectively control viral circulation and the emergence of new variants [4].

Protein subunit vaccines offer a proven and effective approach to advancing global vaccine equity, particularly in low- and middle-income countries (LMICs), where cold-chain constraints and cost-effectiveness are critical considerations. Moreover, their well-established manufacturing processes enable scalable local production, reducing dependence on high-income nations. During the COVID-19 pandemic, several protein subunit vaccines were successfully deployed, demonstrating both high efficacy and suitability for use in resource-limited settings.

SOBERANA^®^ 02 and SOBERANA^®^ Plus are first-generation RBD-based vaccines administered in a heterologous three-dose scheme. Clinical development in both adults and pediatric populations has confirmed its excellent safety profile, as well as strong immunogenicity and efficacy [5,6,7,8]. In particular, a phase I/II clinical trial conducted in children and adolescents aged 3–18 years showed a robust neutralizing and cellular immune response, including the induction of immunological memory, with only 2.6% of participants reporting systemic adverse events [8,9]. Both humoral and cellular immune responses following primary immunization persisted for at least 5–7 months, with some cross-neutralizing immunity observed against the BA.1 Omicron sub-variant [10]. In line with clinical trial results, post-authorization mass vaccination in Cuba showed 83.5% (95% CI: 82.8–84.2) effectiveness against Omicron-related symptomatic disease and 94.6% (95% CI: 82.0–98.6) against severe disease in children aged 3–11 years, with protection sustained over the six months follow-up period [11].

The emergence of new Omicron sub-variants challenges the ability of licensed vaccines to induce specific neutralizing responses, prompting the development of updated mRNA vaccines [12]. However, with only a few manufacturers capable of rapidly developing updated vaccines as new sub-variants emerge, high prices and vaccine shortages have limited immunization—particularly among children—and hindered booster administration [13]. An alternative and less-explored strategy is to evaluate the neutralizing capacity against new sub-variants conferred by first-generation COVID-19 vaccines given as boosters in previously vaccinated individuals.

The present study investigates the safety and cross-neutralizing immunity against Omicron sub-variants among children and adolescents boosted with a single dose of SOBERANA^®^ Plus vaccine applied at least six months after completing the three-dose heterologous primary schedule with SOBERANA^®^ 02 and SOBERANA^®^ Plus vaccines.

## 2. Materials and Methods

### 2.1. Study Design and Approvals

An open-label, multicenter, adaptive phase I/II clinical trial (https://rpcec.sld.cu/trials/RPCEC00000374-En, accessed on 14 March 2025) was conducted to evaluate safety, reactogenicity, and immunogenicity of a primary three-dose vaccination regimen in children aged 3–18 years (stage 1). The regimen consisted of two doses of SOBERANA^®^ 02 (25 µg, SARS-CoV-2 recombinant receptor binding domain [RBD] chemically conjugated to tetanus toxoid and adjuvanted with alum hydroxide) followed by one dose of SOBERANA^®^ Plus (50 µg recombinant dimeric RBD adjuvanted with alum hydroxide), administered 28 days apart. Both vaccines were produced at Finlay Institute of Vaccines in Havana, Cuba. Results of stage 1 have been already published [8].

A modification of the original study protocol, which included follow-up for safety and immunogenicity after primary immunization as well as the application of a booster dose of SOBERANA^®^ Plus (herein Stage 2), was approved by the Ethics Committee of the “Juan Manuel Marquez” Pediatric Hospital and the National Regulatory Agency CECMED (Authorization Reference: 06.004.22BM). Participants received the booster dose a minimum of six months following completion of the three-dose heterologous vaccination series

### 2.2. Subjects and Ethics

For Stage 2, a new written informed consent was obtained from the parents or legal guardians, and informed assent was also obtained from the adolescents (participants aged 12–18 years). Participants with a confirmed RT-PCR diagnosis of SARS-CoV-2 infection during the follow-up period were excluded from receiving the booster dose of SOBERANA^®^ Plus.

Stages 1 and 2 of the study were conducted following the Declaration of Helsinki (Fortaleza, 13 October 2013), Good Clinical Practices and the guidelines of the Cuban National Immunization Program.

### 2.3. Safety Assessment

Following the administration of the booster dose, safety was evaluated by active surveillance by the pediatricians for one hour after vaccination, and during medical visits planned at 24 h, and on day 28. In addition, adverse events were registered by the parents on a daily card until medical visit on day 28.

### 2.4. Immunogenicity Assessment

Immunogenicity assessment was carried out by (a) quantitative ELISA to detect anti-RBD IgG antibodies, (b) molecular virus neutralization titer (mVNT_50_) by competitive ELISA, (c) conventional virus neutralization titer (cVNT_50_) vs. D614G and Omicron (BA.1. 21K) variants; and (d) pseudovirus-based neutralization assay (PBNA) vs. D614G, Omicron EG.5.1 and Omicron XBB.1.5. Procedures for immunologic techniques (a–c) were as previously described [8] and detailed in Appendix A.

Blood samples were collected from all participants prior to and 28 days following the booster dose. To identify prior asymptomatic SARS-CoV-2 infections, pre-booster serum samples were analyzed by ELISA for SARS-CoV-2 nucleocapsid (N) protein. Based on serostatus, the study population was stratified into N-positive and N-negative subgroups for subsequent analyses. A subset of 17 serum samples (28%) from children classified as N-negative before boosting (*n* = 61) were selected using simple random sampling for pseudovirus-based neutralization assay (PBNA).

PBNA assay was performed, adapted from protocol described in reference [14]. Briefly, 200 TCID_50_ of pseudovirus (bearing SARS-CoV-2 Spike variants D614G, Omicron EG.5.1 or Omicron XBB.1.5) were preincubated in cell culture plates (Nunc, Thermo Fisher Scientific, Roskilde, Denmark) with heat-inactivated samples (dilutions 1:60 to 1:14,580) for one hour at 37 °C. Then, 2 × 10^4^ HEK293T/hACE2 cells (Cat#C-HA101, Integral Molecular, Philadelphia, PA, USA) treated with DEAE-Dextran (Cat#D9885-100G, Sigma Aldrich, San Luis, MI, USA) were added. Results were read after 48 h using the EnSight Multimode Plate Reader and BriteLite Plus Luciferase reagent (Cat#6066769, PerkinElmer, Shelton, CT, USA). IC_50_ values were defined as the sample dilution inducing a 50% reduction of viral infectivity and were calculated by non-linear regression of inhibition curves using the GraphPad Prism Software v8.4.3.

The presence of SARS-CoV-2 nucleocapsid (N) protein in serum samples was assessed using the UMELISA SARS-CoV-2 N Protein assay (Immunoassay Center, Havana, Cuba). Briefly, serum samples were diluted 1:20 in assay buffer (0.371 mol/L Tris, 5% sheep serum), and 10 µL of each diluted sample was transferred to ELISA microplates (Greiner Bio-One, Frickenhausen, Germany) pre-coated with recombinant SARS-CoV-2 N-protein (Center for Genetic Engineering and Biotechnology [CIGB], Havana, Cuba). The plates were incubated at 37 °C for 30 min in a humidified chamber. Following incubation, plates were washed, and 10 µL of anti-human IgG antibody conjugated to alkaline phosphatase, diluted in detection buffer (0.05 mol/L Tris, 0.05% Tween 20, 1% bovine serum albumin) was added to each well. Plates were incubated under identical conditions (30 min at 37 °C in a humidified chamber), followed by another wash step. Following this step, 10 µL of 4-methylumbelliferyl phosphate (MUP) substrate was dispensed into each well, and plates were incubated in the dark for 30 min. Fluorescence was measured using a SUMA fluorometric reader (Immunoassay Center, Havana, Cuba). Samples exhibiting fluorescence values exceeding a threshold of 30 relative fluorescence units (RFU) were classified as positive. This cutoff was established as the mean fluorescence of negative control samples plus three standard deviations.

### 2.5. Data Management and Statistical Analysis

Data were collected and managed electronically using the OpenClinica electronic data capture software, (OpenClinica Community Edition version 3.3, Waltham, MA, USA). Safety and reactogenicity data were summarized by reporting frequencies and corresponding percentages. Anti–receptor-binding domain (RBD) IgG concentrations were reported as medians with corresponding interquartile ranges (IQRs). Neutralization titers—both molecular (mVNT_50_) and conventional (cVNT_50_)—were expressed as geometric mean titers (GMTs) with 95% confidence intervals (CIs). Statistical comparisons of paired pre- and post-intervention measurements were performed using the Wilcoxon signed-rank test. The Mann–Whitney U test was used to assess differences between SARS-CoV-2 nucleocapsid (N)-seropositive and N-seronegative individuals. For the pseudovirus-based neutralization assay (PBNA), statistical differences were evaluated using a paired Student’s *t*-test applied to log-transformed neutralization titers to satisfy normality assumptions.

All statistical analyses were conducted using SPSS version 25.0 (IBM Corp., Armonk, NY, USA), EPIDAT version 12.0 (Dirección Xeral de Saúde Pública, Santiago de Compostela, Spain), and GraphPad Prism version 6.0 (GraphPad Software, San Diego, CA, USA). A two-sided *p*-value < 0.05 was considered statistically significant.

## 3. Results

### 3.1. Demographic Characteristic of Subjects and Flow Chart

Following completion of the primary three-dose vaccination series in Stage 1, 306 children were invited to participate in Stage 2 of the trial, which involved a SOBERANA^®^ Plus booster dose administered between February and April 2022. Of these, 62 were excluded: 43 tested positive for SARS-CoV-2 by RT-PCR, 17 lacked parental consent, and 2 had received an alternative COVID-19 vaccine. The remaining 244 children were enrolled in Stage 2 of the study. Demographic characteristics of this cohort are presented in Table 1. A total of 217 subjects were included in the immunogenicity analysis, with pre- and post-booster results and also protein N determination (Figure 1).

### 3.2. Safety After Applying the Booster Dose with SOBERANA^®^ Plus

The participants (244 children) received a SOBERANA^®^ Plus booster dose at least six months after the third dose of the heterologous primary vaccination series (in some children it was administered at seven months). Safety was assessed for 28 days after the booster dose in all participant. The frequency of children with adverse events (AEs) after the booster dose was 18%, with no differences between the 3–11 and 12–18-year age groups. No serious or severe AEs were reported. Local events (85.2%) predominated over systemic events (14.8%) (Table 2). A proportion of 97.7% of AEs were considered mild, and all recovered spontaneously (Appendix A).

Local pain was the most common AE (10.7%), followed by swelling (6.6%), erythema (6.1%), and local warm (5.7%). Fever occurred in less than 1% of subjects (Appendix A). Nine children experienced unsolicited AEs (headache, functional impotence of the arm, and injection site pruritus) that were considered vaccine-related (Appendix A).

### 3.3. Immune Response After Applying the Booster Dose with SOBERANA^®^ Plus

The immune response was assessed before and 28 days after the booster dose of SOBERANA^®^ Plus in children aged 3–18 years. Children with immunological evaluations (n = 217) were grouped as N negative (*n* = 61) or N positive (*n* = 156) according to the detection of N protein in the serum before the booster dose. The children with previous asymptomatic infection (N-positive) had significantly higher IgG levels and neutralization titers than the N-negative children before the booster doses (Figure 2, Appendix A).

Comparison of the immune response before and after the booster dose for paired samples showed a significant increase after vaccination for all variables, in both 3–11 and 12–18 age groups and in both N-negative and N-positive subjects (Figure 2).

Anti-RBD IgG increased significantly from 87.4 UA/mL (CI 95% 37.4; 178.5) to 1062.4 AU/mL (CI 95% 421.6; 1867.2) in N-negative children aged 3–18 years, whereas in N-positive children the increase was from 231.6 AU/mL (CI 106.9; 489.8) to 791.0 (CI 488.2; 1465.6) (Appendix A). By age subgroup, there was no significant difference in IgG response after booster dose in children with or without previous asymptomatic infection (Figure 2A).

The neutralizing activity of the antibody was determined by molecular (mVNT_50_) and conventional (cVNT_50_) neutralization assays using the D614G and Omicron BA.1 virus strain in a subset of samples. The mVNT_50_ and cVNT_50_ against D614G increased significantly after the booster doses in children aged 3–11 and 12–18 years, even if they were N-positive or N-negative (Figure 2B,C). In the global population of children aged 3–18 years and N-protein negative, the cVNT_50_ against D614G resulted in a GMT of 1487.4 (CI 95% 1142.5; 1936.5) and against the Omicron BA.1 variant 1141.4 (CI 95% 721.9; 1804.7) (Figure 2D; Appendix A). Due to the small sample size, it was not possible to compare cVNT_50_ with Omicron BA.1 variant in N-positive children aged 3–11 years. When comparing N-positive and N-negative children, no statistical differences (*p* > 0.05) were found for all immunological variables, except for cVNT_50_ against D614G in children aged 3–11 y/o, where N-positive had higher response (*p* = 0.002).

Deepening into the cross-response against evolving Omicron subvariants, 17 post-booster random samples from N negative subjects were tested by pseudovirus-based neutralization assay. As observed in Figure 3, a booster dose of SOBERANA^®^ Plus elicited neutralization titers against Omicron subvariants EG.5.1 (GMT of 233.9, CI 95% 160.1; 341.5) and XBB.1.5 (GMT of 238.3, CI 95% 148.7; 381.8), although both were significantly lower (*p* < 0.0001) compared with those elicited against D614G.

## 4. Discussion

This study describes the safety and the immune response in children who received a booster dose of first-generation protein subunit vaccine SOBERANA^®^ Plus at least six months after completing the primary vaccination series with the SOBERANA^®^ vaccination schedule.

An excellent safety profile was observed following administration of the booster. The frequency of local pain (10.7%) was lower after the booster than after the last dose of the primary immunization series (47.7%) [8]. Other AEs—such as swelling (6.6%), local warm (5.7%), and erythema (6.1%)—were reported at slightly higher frequencies after booster compared to the third dose of primary series (3.1%, 1.1%, 1.7%, respectively) [8]. Fever, induration and low-grade fever were each reported at a frequency below 2%.

In an open-label study evaluating safety and immune response of ancestral RBD-based vaccine (IndoVac^®^) as a booster dose in 150 healthy individuals aged 12–17 years who had received complete primary schedule of an inactivated vaccine (last dose applied 6–18 months before booster), the incidence rate of AEs until 28 days after booster was 82.7%; with local pain as the most frequent AEs reported (57.3%), followed by myalgia (40.0%) [15]. In a phase III study with another subunit vaccine, Nuvaxovid, administration of a booster dose approximately nine months after the primary series in adolescents aged 12–17 years resulted in injection site tenderness, headache, fatigue, injection site pain, muscle pain and malaise with reported frequencies ranging from 47% to 72% within seven days after the booster [16]. It should be noted that comparisons of AEs rates across are limited by differences in their adjuvant and antigen composition. Nuvaxovid employs Matrix-M™, and IndoVac^®^ uses a combination of alumina and CpG 1018—adjuvants potentially more reactogenic than the alumina-only formulation used in SOBERANA^®^ Plus. Despite these differences, the data consistently support the established safety of the protein subunit vaccine platform.

Although the humoral response induced by the heterologous schedule of SOBERANA^®^ vaccine regimen declined with time [10], administration of a SOBERANA^®^ Plus booster significantly increased all immunological parameters in children aged 3–18 years. Notably, no differences were observed between N-negative and N-positive participants following the booster dose.

Remarkably, a single dose of SOBERANA^®^ Plus, containing RBD from the original SARS-CoV-2 strain, induced high neutralizing antibody titers in N-negative children aged 3–18 years. Compared to pre-booster levels, neutralizing antibodies increased by 24.7-fold against D614G variant and a 55.9-fold against Omicron BA.1 subvariant. This finding aligns with previous observations showing enhanced neutralizing activity against Omicron variant following booster doses of original—strain vaccines. In children aged 5–11 years who received a BNT162b2 booster 7–9 months after completing primary vaccination, neutralizing antibody levels were about 10- and 22-fold higher against ancestral and Omicron (B.1.1.529) variants, respectively [17]. Another study in children and adolescents 3–17 y/o who received a booster dose of inactivated CoronaVac vaccine, 10–12 months after primary two-dose series, reported 28-to 47-fold and 11-to 20-fold increases in neutralizing antibodies against prototype and Omicron strains, respectively, depending on dose and schedule. However, neutralizing titers against the prototype strain were 18–25 times higher than those against Omicron [18]. RBD-based protein subunit boosters have also demonstrated cross-neutralizing responses against Omicron. In adolescents aged 12–17 years, a booster dose of IndoVac^®^—formulated with RBD from the original Wuhan variant—induced a geometric mean fold rise (GMFR) of 8.77 (95% CI: 7.015–10.958) in neutralizing antibodies against Omicron (subvariant not specified) 14 days post-vaccination [15]. Similarly, a single booster dose of CORBEVAX^TM^, another Wuhan RBD-based subunit vaccine, induced a GMFR of 3.033 (95% CI: 2.340–3.932) against the more immune-evasive XBB.1.5 subvariant 28 days after administration in individuals aged 5–80 years [19].

The SARS-CoV-2 virus is undergoing continuous evolution, with globally disseminated Omicron subvariants acquiring mutations in the RBD and spike protein that confer antibody escape [20], making it more difficult for first-generation vaccines to effectively neutralize them [21]. In the previously mentioned CORBEVAX™ trial, the comparison between the original Wuhan RBD-based vaccine and the updated XBB.1.5 RBD-based vaccine showed that the updated formulation induced a higher GMFR in neutralizing antibodies against the XBB.1.5 subvariant: 7.637 (95% CI: 6.090–9.578) versus 3.033 (95% CI: 2.340–3.932) on Day 28 post-booster [19]. Another study with individuals aged 3–83 years who received up to four doses of inactivated, protein-subunit, or recombinant vectored vaccines showed that, in infection-naive individuals, the application of three or four doses led to a significant increase of neutralizing antibodies against ancestral strain and Omicron subvariants BF.7, BQ.1, BQ.1.1 XBB.1, and XBB.1.5,—although responses against XBB.1, and XBB.1.5 were weaker [22]. In our study, a booster dose with ancestral RBD-based SOBERANA^®^ Plus vaccine showed a neutralizing capacity against EC.5.1 and XBB.1.5 subvariants in individuals without prior SARS-CoV-2 infection (N-negative), albeit at lower levels compared to D614G.

The findings of this study support a potential role for SOBERANA^®^ Plus as a heterologous booster in global immunization strategies, including settings where primary vaccination was conducted with non-protein platforms—such as inactivated virus vaccines, as was previously observed in adults [23]. Importantly, despite being based on the ancestral RBD, SOBERANA^®^ Plus elicited substantial cross-neutralizing responses against Omicron subvariants, suggesting residual protective capacity against severe outcomes even in the face of antigenic drift. While neutralizing titers may wane over time and are generally lower against newer Omicron lineages compared to the ancestral strain, current evidence indicates that cross-reactive immunity induced by ancestral-strain vaccines could still contribute meaningfully to protection against severe disease and death. In this context, and in alignment with the WHO’s roadmap on the use of COVID-19 vaccines amid widespread Omicron circulation and high population immunity, vaccines based on the ancestral strain remains as a valid option for booster administration when updated vaccines are unavailable [24].

This study has some limitations. First, the immune response was assessed solely at 28 days after booster administration, as long-term follow-up was not part of the study design. Second, T-cell responses were not evaluated. Another limitation—previously discussed in the publication of the Stage 1 of the trial [8]—was the absence of placebo or control group. Stage 1 was conducted during the Delta variant wave in Cuba (weeks 22–42, 2021), a period marked by a sharp rise in COVID-19 morbidity and mortality across all age groups, including children. Under those circumstances, conducting a placebo-controlled clinical trial was considered unethical.

## 5. Conclusions

In conclusion, administration of a SOBERANA^®^ Plus booster dose elicited a substantial increase in both total and neutralizing antibodies against both D614G and Omicron BA.1 with no safety concerns. Neutralizing activity was also detected against the EG.5.1 and XBB.1.5 Omicron subvariants. Although updated vaccines are recommended to better match circulating subvariants, our findings underscore the value of first-generation RBD-based vaccines as booster doses in children and adolescents, maintaining the recognition of new variants of virus with an excellent safety profile.

## Figures and Tables

**Figure 1 vaccines-13-01198-f001:**
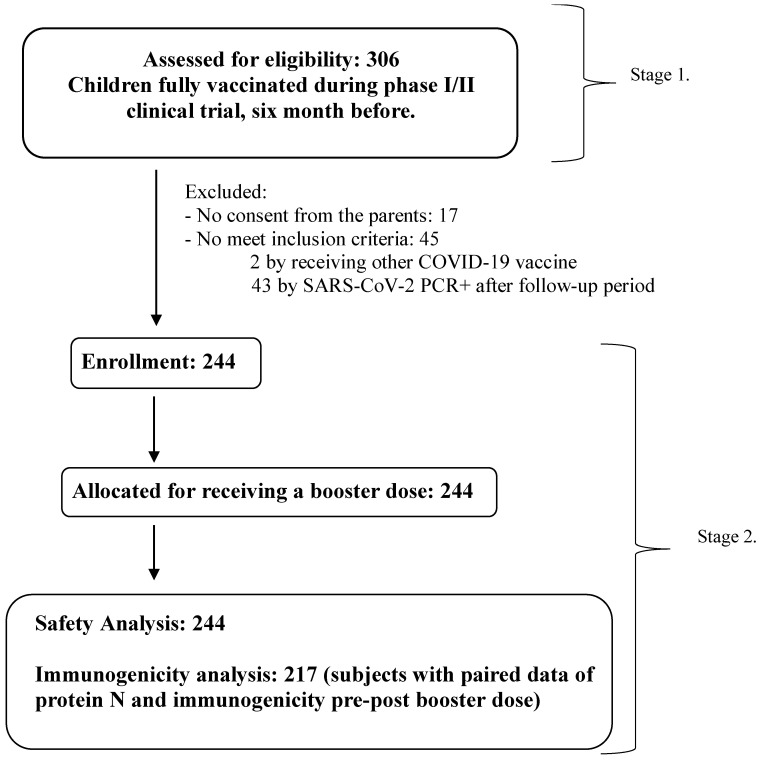
Flow chart of phase I/II clinical trial in children aged 3–18 years. Stage 1: Primary immunization with two doses of SOBERANA^®^02 and a heterologous third dose of SOBERANA^®^ Plus (28 days apart), published in reference [8]. Stage 2: Booster dose with SOBERANA^®^ Plus administered at least 6 months after the third dose of the heterologous scheme.

**Figure 2 vaccines-13-01198-f002:**
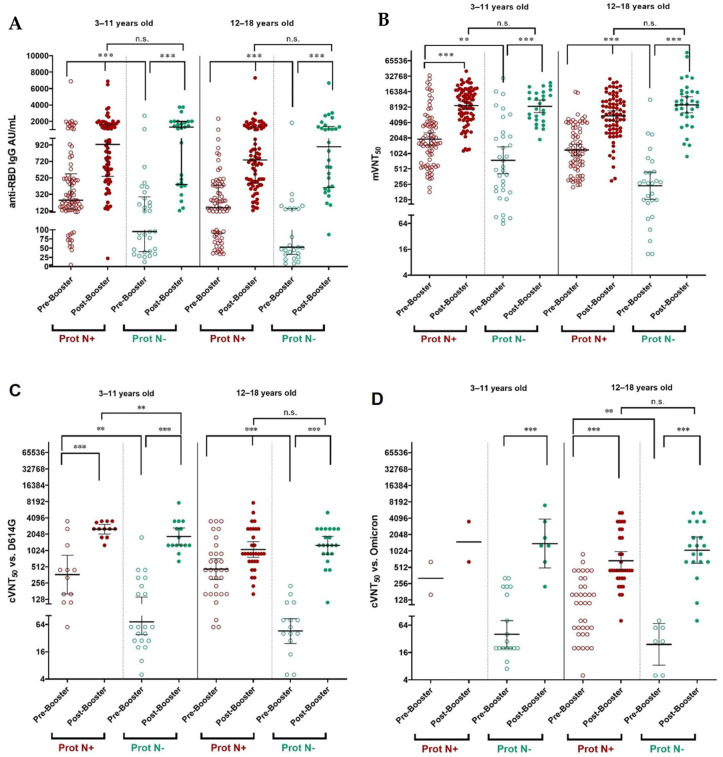
Antibody response before and 28 days after a booster dose with SOBERANA^®^ Plus, in protein N-negative (Prot N−) and positive (Prot N+) children aged 3–11 and 12–18 y/o. (**A**) Anti-RBD IgG concentration expressed in AU/mL (median, 25th–75th percentile) (Prot N−; *n* = 61/Prot N+; *n* = 156). (**B**) Molecular virus neutralization titer mVNT_50_, highest serum dilution inhibiting 50% of RBD:hACE2 interaction GMT [95% CI] (Prot N−; *n* = 61/Prot N+; *n* = 156). (**C**) Conventional live-virus neutralization titer cVNT_50_ GMT [95% CI] against SARS-CoV-2 D614G variant (Prot N−; *n* = 36/Prot N+; *n* = 43); (**D**) Conventional live-virus neutralization titer cVNT_50_ GMT [95% CI] against SARS-CoV-2 Omicron BA.1 variant (Prot N−; *n* = 26/Prot N+; *n* = 26). Statistics: Wilcoxon Signed-Rank Test (before-after booster) and Mann-Whitney U test (Prot N+ vs. Prot N− subjects); ** *p* < 0.005, *** *p* < 0.0005. ns, not significant.

**Figure 3 vaccines-13-01198-f003:**
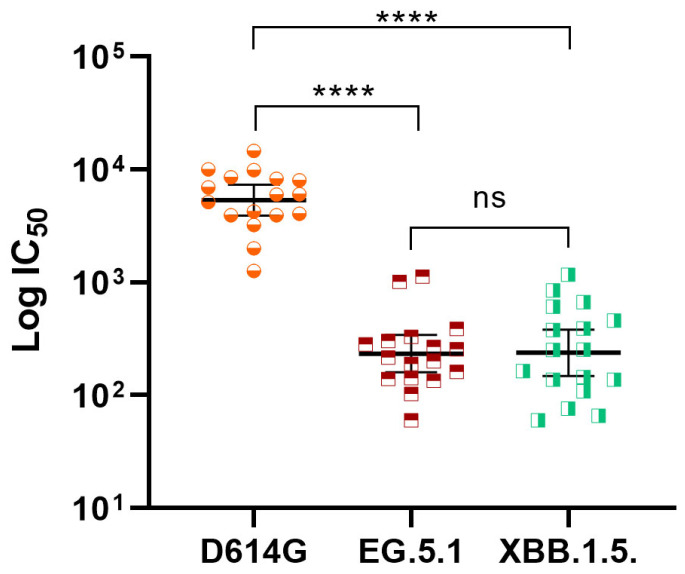
Neutralization titer against SARS-CoV-2 variants by pseudovirus-based neutralization assay in 17 subjects aged 3–18 y/o after receiving a booster dose with SOBERANA^®^ Plus. Sera samples were collected 28 days after booster and evaluated (CI_50_: GMT [95% CI]) against pseudovirus bearing SARS-CoV-2 Spike variants D614G, Omicron EG.5.1 or Omicron XBB.1.5. Only serum samples negative for SARS-CoV-2 nucleocapsid (N) protein by ELISA were selected for this test. Paired Student *t* test with log-transformed variables was used for statistic comparisons: **** *p* < 0.001, ns, not significant.

**Table 1 vaccines-13-01198-t001:** Demographic characteristics of subjects included in the clinical trial.

	Age Groups		
	3–11 Years	12–18 Years	Total
** *n* **	129	115	244
**Sex**			
Female	58 (45.0%)	52 (45.2%)	110 (45.1%)
Male	71 (55.0%)	63 (54.8%)	134 (54.9%)
**Skin color**			
White	93 (72.1%)	73 (63.5%)	166 (68.0%)
Black	8 (6.2%)	10 (8.7%)	18 (7.4%)
Multiracial	28 (21.7%)	32 (27.8%)	60 (24.6%)
**Age (years)**			
Mean (SD)	7.5 (2.5)	14.8 (2.1)	10.9 (4.4)
Median (IQR)	8.0 (5.0)	15.0 (4.0)	11.0 (6.0)
Range	3–11	12–18	3–18
**Weight (kg)**			
Mean (SD)	29.2 (9.9)	54.1 (9.0)	40.9 (15.6)
Median (IQR)	27.5 (14.0)	55.0 (12.0)	42.0 (27.0)
Range	14.0–55.0	32.0–73.0	14.0–73.0
**Height (cm)**			
Mean (SD)	128.9 (17.2)	164.2 (9.8)	145.5 (22.6)
Median (IQR)	131.0 (26.0)	163.0 (14.0)	150.0 (33.0)
Range	94–169	142–190	94–190
**BMI (kg/m^2^)**			
Mean (SD)	17.0 (2.0)	19.9 (2.3)	18.4 (2.6)
Median (IQR)	16.7 (2.7)	19.8 (3.9)	18.1 (3.7)
Range	13.6–22.8	14.6–24.6	13.6–24.6

Data are n (%) unless otherwise specified. Mean (SD) = Mean ± Standard Deviation. Median (IQR) = Median ± Interquartile Range. BMI = Body mass index. Range = (Minimum; Maximum).

**Table 2 vaccines-13-01198-t002:** Frequency and general characteristics of adverse events after the booster dose with SOBERANA^®^ Plus.

	Age Group		
	3–11 y/o	12–18 y/o	Total
** *n* **	129	115	244
Subjects with some AE	22 (17.1%)	22 (19.1%)	44 (18.0%)
Subjects with some vaccine-related AE	20 (15.5%)	21 (18.3%)	41 (16.8%)
Subjects with some serious AE	0 (0.0%)	0 (0.0%)	0 (0.0%)
Subjects with some vaccine-related serious AE	0 (0.0%)	0 (0.0%)	0 (0.0%)
Subjects with some severe AE	0 (0.0%)	0 (0.0%)	0 (0.0%)
Subjects with some vaccine-related severe AE	0 (0.0%)	0 (0.0%)	0 (0.0%)
**Total of AE**	50	38	88
Mild	49 (98.0%)	37 (97.4%)	86 (97.7%)
Moderate	1 (2.0%)	1 (2.6%)	2 (2.3%)
**Serious AE**	0 (0.0%)	0 (0.0%)	0 (0.0%)
Local	42 (84.0%)	33 (86.8%)	75 (85.2%)
Systemic	8 (16.0%)	5 (13.2%)	13 (14.8%)
**AE consistent with vaccination**	46 (92.0%)	36 (94.7)	82 (93.2%)
Serious AE consistent	0 (0.0%)	0 (0.0%)	0 (0.0%)
Severe AE consistent	0 (0.0%)	0 (0.0%)	0 (0.0%)

AE = Adverse Event.

## Data Availability

Data are available upon request.

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
