# Peer review of "Safety and Cross-Neutralizing Immunity Against SARS-CoV-2 Omicron Sub-Variant After a Booster Dose with SOBERANA^®^ Plus in Children and Adolescents"

_vaccines, 2025, doi:10.3390/vaccines13121198_

Round 1

Reviewer 1 Report

Comments and Suggestions for Authors

Article

Title: Safety and cross-reactive immunity against SARS-CoV-2 Omicron sub-variant after a booster dose with SOBERANA® Plus in children and adolescents

General Evaluation

This manuscript addresses an important and timely question: the safety and cross-reactive immunogenicity of the SOBERANA® Plus vaccine booster in children and adolescents. The topic is relevant for global vaccination strategies, especially in resource-limited settings where access to updated mRNA vaccines remains challenging. The study is well designed, presents valuable data, and is written with overall clarity. The findings contribute to the understanding of first-generation RBD-based vaccine performance against emerging Omicron subvariants. However, several editorial, methodological, and presentation-related issues require revision to enhance the scientific rigor and overall quality of the manuscript.

  1. Title and Abstract

Positive Aspects:

  • The title accurately conveys the study’s scope and population.
  • The abstract is concise, informative, and structured according to the journal’s format.
  • The background appropriately contextualizes the study within the ongoing evolution of SARS-CoV-2 variants.

Required Revisions:

  • Ensure consistent use of decimal points (replace commas with points in all numerical data, e.g., “10,7%” → “10.7%”).
  • Include numerical data on key immunogenicity outcomes directly in the abstract to provide quantitative context.
  • Minor stylistic improvement: replace “cross-reactive immunity” with “cross-neutralizing immunity” for precision.
  • Ensure trial registry number and ethical approval are mentioned within the abstract or keywords, in line with MDPI guidelines.

  1. Introduction

Positive Aspects:

  • The introduction is well written and establishes the clinical relevance of booster vaccination in the pediatric population.
  • The rationale for testing SOBERANA® Plus as a booster is clear and appropriately referenced.

Required Revisions:

  • The introduction would benefit from a short paragraph linking this work to global vaccine equity and the role of subunit vaccines in low- and middle-income countries.
  • Consider citing additional international studies evaluating subunit boosters in children (for example, Novavax or recombinant RBD studies).
  • Verify consistency and update references to ensure they are the most recent (WHO and CDC updates for 2025).

  1. Materials and Methods

Positive Aspects:

  • The methods are described in detail, including the ethical framework, assays used, and analytical procedures.
  • The use of pseudovirus-based neutralization assays and molecular virus neutralization titers (mVNT50) is appropriate and robust.

Required Revisions:

  • Decimal notation correction: Throughout the methods section, replace commas with decimal points (e.g., “0,05 mol/L” → “0.05 mol/L”).
  • Clarify whether the pseudovirus assays for Omicron XBB.1.5 and EG.5.1 were validated or adapted from previous protocols.
  • Specify sample size justification or power analysis, especially for the subset (n = 17) used for pseudovirus neutralization.
  • Indicate the manufacturer, country, and catalog number for all major reagents and instruments (e.g., ELISA kits, luciferase reagents).
  • Clarify the rationale for selecting only N-negative participants for certain assays and whether this introduces selection bias.

  1. Results

Positive Aspects:

  • The results are well organized, progressing logically from safety to immunogenicity.
  • Data presentation in Tables 1–2 and Figures 1–3 is clear and supports the conclusions.
  • Statistical analysis methods are appropriate and well explained.

Required Revisions:

  • Decimal consistency: Replace commas with points in all numerical values, including tables and figure captions.
  • Figures should be checked for resolution, axis labeling, and font uniformity—the font size in Figure 2 appears inconsistent and may not meet MDPI resolution requirements.
  • In Table 1, replace semicolons used as decimal or range separators with standard format (e.g., “Range 3;11” → “Range 3–11”).
  • Clarify in Table 2 whether the total number of adverse events (AEs) corresponds to event count or number of affected subjects.
  • Ensure that supplementary tables (S1–S4) are referenced sequentially and correspond exactly to the order described in the main text.

  1. Discussion

Positive Aspects:

  • The discussion is balanced and compares the findings with previous studies effectively.
  • The authors appropriately acknowledge limitations, including the absence of T-cell response data and long-term follow-up.

Required Revisions:

  • Strengthen the interpretation of cross-neutralization findings by discussing implications for protection against newer Omicron lineages and potential waning immunity.
  • Discuss whether the observed cross-reactive antibody titers correlate with clinical protection—even speculative discussion would enhance relevance.
  • Add a brief paragraph on the potential role of SOBERANA® Plus in heterologous booster strategiesinternationally, especially in countries where primary immunization used other vaccine platforms.
  • The discussion could benefit from a more explicit connection to global pediatric vaccination policies (e.g., WHO SAGE recommendations).

  1. Tables and Figures

Positive Aspects:

  • Figures are informative and visually represent immunological data effectively.
  • Flowchart (Figure 1) provides clear understanding of study stages and participant selection.

Required Revisions:

  • Ensure uniform graphical style (axis scales, legend placement, and color schemes).
  • Increase image resolution to meet journal requirements (minimum 300 dpi).
  • Figures 2 and 3 should include exact sample sizes in the legends.
  • Replace all commas with decimal points in numeric data and figure annotations.
  • Verify alignment of table headers and footnotes, ensuring proper reference to units (e.g., “AU/mL,” “GMT [95% CI]”).

  1. References

Positive Aspects:

  • The reference list is comprehensive and relevant, citing both Cuban and international literature.
  • Recent publications (2023–2025) are included.

Required Revisions:

  • Verify full compliance with MDPI style (use of full journal names, correct DOI format, spacing).
  • Some references are missing final access dates for URLs (e.g., WHO, Scientific American); include exact “accessed on” dates.
  • Ensure that all references are correctly numbered and cited in the text in sequential order.
  • Suggest including additional comparative studies of pediatric boosters using other platforms to reinforce the external validity of findings.

  1. Language and Formatting

Positive Aspects:

  • The manuscript is well written and readable.
  • Terminology is scientifically accurate and consistent.

Required Revisions:

  • Carefully review for minor grammatical and stylistic inconsistencies (e.g., “children ś parents” should be “children’s parents”).
  • Perform a final English language revision by a native or professional editor, as minor syntactic issues persist.
  • Correct all numerical punctuation: commas must be replaced with points in every numeric expression to comply with international scientific formatting.

  1. Ethical and Transparency Statements

Positive Aspects:

  • Ethical approvals and informed consent processes are well described.
  • Conflicts of interest are transparently disclosed.

Required Revisions:

  • Explicitly state compliance with the Declaration of Helsinki (latest version, 2013) and local regulations in the Methods section.
  • Include confirmation that the trial adhered to Good Clinical Practice (GCP) standards.

Overall Recommendation:

This manuscript presents high-quality and relevant clinical data on pediatric COVID-19 vaccination; however, it requires minor editorial and formatting revisions, particularly:

  • Correction of decimal punctuation (comma → point) throughout the text, tables, and figures.
  • Minor methodological clarifications (subset selection, assay validation).
  • Enhanced discussion of global context and implications.

After these corrections and a thorough English language review, the manuscript would be suitable for publication in Vaccines.

Author Response

2. Point-by-point response to Comments and Suggestions for Authors

Title and Abstract:

Comments 1: Ensure consistent use of decimal points (replace commas with points in all numerical data, e.g., “10,7%” 10.7%).

Response 1: Thank you for pointing this out. We have corrected that issue in the whole manuscript (text, supplementary file, tables and figures)

Comments 2: Include numerical data on key immunogenicity outcomes directly in the abstract to provide quantitative context.

Response 2: Agree. We have included in the abstract the fold increase of the neutralizing antibodies vs D614G and Omicron after application of the booster dose (page 1, line 39) as follows: “All humoral response parameters increased significantly post-booster, including neutralizing antibodies against D614G (24.7-fold increase) and Omicron BA.1 (55.9-fold increase), with similar responses in N-negative and N-positive individuals.”

Comments 3: Minor stylistic improvement: replace “cross-reactive immunity” with “cross-neutralizing immunity” for precision.

Response 3: Agree. We have replaced “cross-reactive immunity” with “cross-neutralizing immunity” in title, keywords and introduction following your suggestion.

Comments 4: Ensure trial registry number and ethical approval are mentioned within the abstract or keywords, in line with MDPI guidelines.

Response 4: Agree. Trial registry number and ethical approval code are mentioned at the end of the abstract section (page 2, lines 46-47)

Introduction:

Comments 1: The introduction would benefit from a short paragraph linking this work to global vaccine equity and the role of subunit vaccines in low- and middle-income countries

Response 1: Agree. We have added the following paragraph in the introduction section (page 2, lines 59-65: “Protein subunit vaccines offer a proven and effective approach to advancing global vaccine equity, particularly in low- and middle-income countries (LMICs), where cold-chain constraints and cost-effectiveness are critical considerations. Moreover, their well-established manufacturing processes enable scalable local production, reducing dependence on high-income nations. During the COVID-19 pandemic, several protein subunit vaccines were successfully deployed, demonstrating both high efficacy and suitability for use in resource-limited settings.

Comments 2: Consider citing additional international studies evaluating subunit boosters in children (for example, Novavax or recombinant RBD studies).

Response 2: We acknowledge the relevance of these studies; however, to preserve the focus and conciseness of the Introduction, we considered it more appropriate to reserve detailed discussion of international studies evaluating subunit boosters in children for the Discussion section. Following your suggestion, we have included two studies evaluating RBD-based vaccines as boosters in children and adolescents. 

Comments 3: Verify consistency and update references to ensure they are the most recent (WHO and CDC updates for 2025)

Response 3: We have verified the references, particularly reference 2, regarding WHO´s COVID-19 vaccines advice from October 8, 2024, which remains current and valid as of October 2025.

Materials and methods:

Comments 1: Decimal notation correction: Throughout the methods section, replace commas with decimal points (e.g., “0,05 mol/L” 0.05 mol/L).

Response 1: Thank you for pointing this out. We have corrected that issue throughout the methods section

Comments 2: Clarify whether the pseudovirus assays for Omicron XBB.1.5 and EG.5.1 were validated or adapted from previous protocols

Response 2: The pseudovirus assays for Omicron XBB.1.5 and EG.5.1 were adapted from previous protocols. In the current version of manuscript that issue was clarified

Comments 3: Specify sample size justification or power analysis, especially for the subset (n = 17) used for pseudovirus neutralization.

Response 3: A formal sample size calculation was not performed for the subset used for pseudovirus neutralization. Due to laboratory constraints limiting the number of samples that could be analyzed, we deemed a randomly selected subset of 28% (17 of 61 N-protein–negative samples), chosen by simple random sampling, adequate for an initial assessment of the response against Omicron XBB.1.5 and EG.5.1.

Comments 4: Indicate the manufacturer, country, and catalog number for all major reagents and instruments (e.g., ELISA kits, luciferase reagents).

Response 4: Thank you for pointing this out. We have indicated manufacturer, country, and catalog number for all major reagents and instruments in the corrected version of the manuscript.

Comments 5: Clarify the rationale for selecting only N-negative participants for certain assays and whether this introduces selection bias.

Response 5: In our manuscript, we stratified the study population into N-protein–positive and N-protein–negative subgroups to evaluate the influence of prior SARS-CoV-2 infection during the six-month follow-up period on immunogenicity following booster vaccination. Most immunogenicity parameters—specifically anti-RBD IgG, molecular virus neutralization (mVNT₅₀), and conventional virus neutralization (cVNT₅₀)—were analyzed while accounting for N-protein serostatus.

The pseudovirus neutralization assay (PBNA) against the Omicron subvariants XBB.1.5 and EG.5.1, however, was performed only in N-protein–negative participants. These variants emerged several months after the majority of immunogenicity assessments in the clinical trial had been completed. Then, the primary objective of the PBNA was to assess cross-neutralizing immunity induced by the booster against these newly circulating variants in individuals without prior infection. To avoid potential confounding from hybrid immunity (i.e., the combined effect of natural infection and vaccination), samples from N-protein–positive participants were excluded from this specific analysis.

While an analysis in N-protein–positive individuals would certainly be of interest, laboratory capacity constraints limited the total number of samples we could process. Therefore, we prioritized N-protein–negative samples to specifically address the question of vaccine-induced immunity in the absence of prior infection.

Results:

Comments 1: Decimal consistency: Replace commas with points in all numerical values, including tables and figure captions.

Response 1: Thank you for pointing this out. We have corrected that issue in the whole manuscript (text, supplementary file, tables and figures)

Comments 2: Figures should be checked for resolution, axis labeling, and font uniformity—the font size in Figure 2 appears inconsistent and may not meet MDPI resolution requirements.

Response 2: Thank you for pointing this out. We have corrected that issue in all the figures, resolution was fixed in 300 dpi

Comments 3: In Table 1, replace semicolons used as decimal or range separators with standard format (e.g., “Range 3;11” Range 311).

Response 3: Thank you for pointing this out. We have corrected that issue in table 1

Comments 4: Clarify in Table 2 whether the total number of adverse events (AEs) corresponds to event count or number of affected subjects

Response 4: In table 2, row “N” corresponds the number of subjects in each sub-group (3-11 y/o or 12-18 y/o) and total of subjects with safety analysis performed. The row “Subjects with some AE” corresponds to the number of subjects that reported at least one AE, stratified by the two age subgroups and the total number in the last column. The whole table follows the same format.     

Comments 5: Ensure that supplementary tables (S1–S4) are referenced sequentially and correspond exactly to the order described in the main text.

Response 5: Thank you for pointing this out. There was a mistake in previous version because Table S2 was referenced in the text together with Table S1. That issue was corrected in current version of the manuscript.    

Discussion:

Comments 1: Strengthen the interpretation of cross-neutralization findings by discussing implications for protection against newer Omicron lineages and potential waning immunity.

Comments 2: Discuss whether the observed cross-reactive antibody titers correlate with clinical protection—even speculative discussion would enhance relevance.

Comments 3: Add a brief paragraph on the potential role of SOBERANA® Plus in heterologous booster strategies internationally, especially in countries where primary immunization used other vaccine platforms

Comments 4: The discussion could benefit from a more explicit connection to global pediatric vaccination policies (e.g., WHO SAGE recommendations).

Responses 1-4: Thank you for your valuable suggestions. All have been carefully considered, and we have added a new paragraph to the Discussion section (page 12, lines 344–357) addressing the potential role of SOBERANA® Plus in heterologous booster strategies in the context of emerging Omicron subvariants and waning immunity. At the same time, as highlighted in the WHO’s roadmap on the use of COVID-19 vaccines, first-generation vaccines continue to provide meaningful protection against severe disease and death and remain a viable option when updated vaccines are unavailable.

The new paragraph states as follows: “The findings of this study support a potential role for SOBERANA® Plus as a heterologous booster in global immunization strategies, including settings where primary vaccination was conducted with non-protein platforms—such as inactivated virus vaccines, as was previously observed in adults [23]. Importantly, despite being based on the ancestral RBD, SOBERANA® Plus elicited substantial cross-neutralizing responses against Omicron subvariants, suggesting residual protective capacity against severe outcomes even in the face of antigenic drift. While neutralizing titers may wane over time and are generally lower against newer Omicron lineages compared to the ancestral strain, current evidence indicates that cross-reactive immunity induced by ancestral-strain vaccines could still contribute meaningfully to protection against severe disease and death. In this context, and in alignment with the WHO’s roadmap on the use of COVID-19 vaccines amid widespread Omicron circulation and high population immunity, vaccines based on the ancestral strain remains as a valid option for booster administration when updated vaccines are unavailable [24].”

23.     Ramezani, A.; Sorouri, R.; Haji Maghsoudi, S.; Dahmardeh, S.; Doroud, D.; Sadat Larijani, M.; et al. Pasto-Covac and PastoCovac Plus as protein subunit COVID-19 vaccines led to great humoral immune responses in BBIP-CorV immunized individuals. Sci Rep 2023, 13, 8065. https://doi.org/10.1038/s41598-023-35147-y

24.     W.H.O. WHO SAGE roadmap on uses of COVID-19 vaccines in the context of Omicron and high population immunity. November 10, 2023. Available online: https://www.who.int/publications/i/item/WHO-2019-nCoV-Vaccines-SAGE-Prioritization-2023.1 (accessed October 16, 2025).

Tables and Figures:

Comments 1: Ensure uniform graphical style (axis scales, legend placement, and color schemes).

Response 1: Thank you for pointing this out. We have corrected all figure following your suggestions

Comments 2: Increase image resolution to meet journal requirements (minimum 300 dpi).

Response 2: Thank you for pointing this out. We have corrected that issue in all the figures, resolution was fixed in 300 dpi

Comments 3: Figures 2 and 3 should include exact sample sizes in the legends.

Response 3: Thank you for pointing this out. We have added samples size for each test performed in the legend of figure 2 and figure 3. (pages 9 and 10)

Comments 4: Replace all commas with decimal points in numeric data and figure annotations

Response 4: Thank you for pointing this out. We have corrected that issue in numeric data and figure annotations

Comments 5: Verify alignment of table headers and footnotes, ensuring proper reference to units (e.g., “AU/mL,” “GMT [95% CI]”).

Response 5: Thank you for pointing this out. We have corrected that issue in table headers and footnotes

References:

Comments 1: Verify full compliance with MDPI style (use of full journal names, correct DOI format, spacing).

Response 1: All references have been verified and are in compliance with MDPI style. For Vaccines journal is requested abbreviated journal name (Journal Articles: 1. Author 1, A.B.; Author 2, C.D. Title of the article. Abbreviated Journal Name Year, Volume, page range.)

Comments 2: Some references are missing final access dates for URLs (e.g., WHO, Scientific American); include exact “accessed on” dates

Response 2: All references with URLs have their access dates

Comments 3: Ensure that all references are correctly numbered and cited in the text in sequential order

Comments 4: Suggest including additional comparative studies of pediatric boosters using other platforms to reinforce the external validity of findings

Response 4: Following your suggestion, we have included two studies evaluating RBD-based vaccines as boosters in children and adolescents. 

Language and formatting:

Comments 1: Carefully review for minor grammatical and stylistic inconsistencies (e.g., “children ś parents” should be “children’s parents”).

Response 1: The manuscript has been thoroughly reviewed and revised for language and grammatical

Comments 2: Perform a final English language revision by a native or professional editor, as minor syntactic issues persist.

Response 2: The manuscript has been thoroughly reviewed and revised for language and grammatical

Comments 3: Correct all numerical punctuation: commas must be replaced with points in every numeric expression to comply with international scientific formatting

Response 3: We have corrected that issue in the whole manuscript (text, supplementary file, tables and figures)

Ethical and Transparency Statements:

Explicitly state compliance with the Declaration of Helsinki (latest version, 2013) and local regulations in the Methods section. Include confirmation that the trial adhered to Good Clinical Practice (GCP) standards.

Response: Thank you for pointing this out. We have included in section “2.2 Subjects and ethics” a paragraph including both statements as follows: “Stages 1 and 2 of the study were conducted following the Declaration of Helsinki (Fortaleza, 13th Oct, 2013), Good Clinical Practices and the guidelines of the Cuban National Immunization Program.” Page 3, lines 115-117

Reviewer 2 Report

Comments and Suggestions for Authors

Summary

The authors identify the emergence of the SARS-CoV-2 Omicron sub-variants as an ongoing need for booster immunisation across all age groups. They state that the heterologous three-dose regimen of SOBERANA 02 and SOBERANA Plus that are protein subunit vaccines based on the original RBD have proved safe, immunogenic and effective in paediatric populations as a primary series. The purpose of their study was to evaluate the safety and immunogenicity of a SOBERANA Plus booster dose administered six months post primary vaccination in persons 3-18y/o. This was a followed up analysis of a phase1/II trial of 244 participants receiving a Plus booster. They monitored safety using active surveillance at 1 hour, 24 hours and over 28 days post-vaccination. They assessed humoral responses 28 days post-booster. Serum samples were tested for antibodies against the SARS-CoV-2 nucleocapsid (N) protein. 

They observed adverse events in 18% of study participants,; 85.2% local vs 14.8% systemic;no AEs or SAEs were recorded. All humoral responses increased significantly post-booster including neutralising antibodies against D614G and Omicron BA.1, ith similar responses in N-negative ands N-positive individuals. They also detected cross-neutralisation against recent variants (XBB.1.5 and EG.5.1).

They conclude that the Plus booster is safe and significantly enhances cross-reactive NAbs against evolving Omicron sub-variants in children and adolescents, and that RBD-based vaccines have the potential to sustain broad immunity in the face of viral evolution.

General comments

The introduction comprehensively narrates the background of this protein subunit vaccine and the studies done to date. It also provides the rationale for its prioritisation as part of the preventive arsenal against not only SARS-CoV-2, but as a platform for other potential vaccines.  

The materials and methods section provides relevant documentation of prior studies and the consenting and enrolment process, the inclusion and exclusion criteria for the subjects. This is aided by Figure 1. What is not clearly stated is the safety assessments at 24 hours and day 28. Did all subjects return to the study centre or were project staff dispatched to participants’ homes? A sentence can clarify the “professional surveillance”. The immunogenicity assessment is clearly described, as is the data management.  

The results section is well organised, and substantially aided by figures 2 and 3. 

The discussion is organised and succinct. The study limitations include perhaps the most important one, the lack of a placebo or control group in Stage 1. However, given the lack of SAEs in this study, including any attributable to the vaccine, the conclusions are valid. 

Specific comments 

None. 

Author Response

2. Point-by-point response to Comments and Suggestions for Authors

Comments: The materials and methods section provide relevant documentation of prior studies and the consenting and enrolment process, the inclusion and exclusion criteria for the subjects. This is aided by Figure 1. What is not clearly stated is the safety assessments at 24 hours and day 28. Did all subjects return to the study centre or were project staff dispatched to participants’ homes? A sentence can clarify the “professional surveillance”.

Response: Thank you for pointing this out. We have modified the whole paragraph to clarify both aspects as follows: “Following the administration of the booster dose, safety was evaluated by active surveillance by the pediatricians for one hour after vaccination, and during medical visits planned at 24 hours, and on day 28. In addition, adverse events were registered by the parents on a daily card until medical visit on day 28” (see section 2.3 in the corrected version of the manuscript).

Reviewer 3 Report

Comments and Suggestions for Authors

This manuscript reports immunogenicity and reactogenicity results from a phase I/II clinical trial of a SOBERANA® Plus booster dose administered six months after a primary SOBERANA® 02 series in children and adolescents aged 3–18 years. Participants were stratified into two age groups (3–11 and 12–18 years). The study provides data on booster vaccination in paediatric populations, including evidence of cross-reactive immunity against emerging SARS-CoV-2 Omicron subvariants.

The manuscript is well-structured, clearly written, and methodologically sound. The results contribute valuable evidence supporting protein subunit vaccines as booster options in paediatrics.

Comments.
1. Posology (Section 2.1):
Consider specifying the exact vaccine dose administered in this trial. If the paediatric dosage differs from that used in adults, this should be clarified explicitly.

2. Figure 3 (Lines 236–240):
The text description of Figure 3 provides only general trends, whereas Figures 2A–D report geometric mean titres (GMTs). For consistency and clarity, please add numerical values (e.g., GMT with 95% CI) to the body text when describing Figure 3 results.

3. Comparison with other subunit vaccines (Discussion, lines 262–266):
The comparison with Nuvaxovid is somewhat limited. Nuvaxovid uses Matrix-M as an adjuvant, which may enhance both immunogenicity and reactogenicity, whereas SOBERANA employs aluminium-based adjuvant, a classical system with different properties. Consider expanding the discussion by comparing immunological outcomes more thoroughly and highlighting how adjuvant choice may influence reactogenicity and antibody responses.

4. Terminology (Line 43, Abstract/Conclusions)
The manuscript assessed only short-tern immunogenicity at a single post-booster timepoint, without long-term follow-up or efficacy/effectiveness data. 
The phrases such as “sustain broad immune protection” may therefore be overstated. Consider rephrasing to terms such as “stimulate” or “maintain broad immunity,” or similar wording more directly aligned with your findings

Errors and format.
1. Subscripts:
Please use consistent formatting with subscripts throughout (e.g., cVNT₅₀, mVNT₅₀, TCID₅₀, IC₅₀), including figure labels.

2. Units:
Standardise units according to SI conventions. For example, “L” should be used consistently for litre, avoiding variation between “l” and “L” (e.g., μL vs μl; mol/L vs mol/l).

3. Figures 2C and 2D:
Both panels present cVNT₅₀ data but against different virus strains. Standardising the y-axis scale would allow a clearer visual comparison across panels.

4. Capitalisation:
Ensure consistent use of “Omicron” (capitalised as a proper noun) in all figures, tables, and text. For instance, corrections are needed at lines 117, 122, 123, 230, 239, 246, 283, and elsewhere.

5. Decimal separator:
Please use the period (“.”) as the decimal separator instead of the comma (“,”) to align with the journal’s formatting style.

6. Table S3 (Supplementary material)
The three subgroup columns are all labelled “3–11 y/o.” This appears to be an error. They should likely read 3–11 y/o, 12–18 y/o, and Total. Please verify and correct.

Author Response

2. Point-by-point response to Comments and Suggestions for Authors

Comments 1: Posology (Section 2.1): Consider specifying the exact vaccine dose administered in this trial. If the pediatric dosage differs from that used in adults, this should be clarified explicitly.

Response 1: Thank you for pointing this out. We have included the dose of the vaccine in the materials and methods section 2.1, first paragraph.

Regarding the pediatric dosage, for SOBERANA vaccines, the dosage for children ≥ 2 years old is the same as in adults: SOBERANA® 02 (25 µg) and SOBERANA® Plus (50 µg).

Comment from the authors: During vaccine development was designed that way taking into account: 1) the range of antigen content used in other established protein-based vaccines routinely administered to children and even infants. For instance, the tetanus toxoid vaccine contains 40 µg of protein, the diphtheria toxoid vaccine 75 µg, Bexsero® delivers 50 µg of recombinant proteins plus 25 µg of outer membrane vesicles, and Trumenba® uses 60 µg of recombinant antigens; and 2) more importantly, the favorable safety profile of protein subunit vaccines in pediatric populations.

Given these elements we determined that evaluating the same formulation and dosing schedule used in adults was both scientifically justified and ethically appropriate for children. This approach ensures consistency in immune response assessment while leveraging the well-documented tolerability of protein subunit platforms in young populations.

Comments 2: Figure 3 (Lines 236–240): The text description of Figure 3 provides only general trends, whereas Figures 2A–D report geometric mean titres (GMTs). For consistency and clarity, please add numerical values (e.g., GMT with 95% CI) to the body text when describing Figure 3 results.

Response 2: Thank you for pointing this out. We have included numerical numbers of GMT with 95%CI in the body text description of Figure 3 as follows: “As observed in figure 3, a booster dose of SOBERANA® Plus elicited neutralization titers against Omicron subvariants EG.5.1 (GMT of 233.9, CI 95% 160.1; 341.5) and XBB.1.5 (GMT of 238.3, CI 95% 148.7; 381.8), although both were significantly lower (p<0.0001) compared with those elicited against D614G.” (see corrected version of manuscript page 9, lines 265-658.

Comments 3: Comparison with other subunit vaccines (Discussion, lines 262–266): The comparison with Nuvaxovid is somewhat limited. Nuvaxovid uses Matrix-M as an adjuvant, which may enhance both immunogenicity and reactogenicity, whereas SOBERANA employs aluminum-based adjuvant, a classical system with different properties. Consider expanding the discussion by comparing immunological outcomes more thoroughly and highlighting how adjuvant choice may influence reactogenicity and antibody responses.

Response 3: Thank you for pointing this out. In the revised manuscript, we have included an additional vaccine that is more similar to SOBERANA® in terms of composition. We have also added a discussion of the limitations associated with comparing reactogenicity and immunogenicity across vaccines that differ in adjuvant and/or antigen composition. The revised paragraph (page 11, lines 287–301) now reads as follows: “In an open-label study evaluating safety and immune response of ancestral RBD-based vaccine (IndoVac®) as a booster dose in 150 healthy individuals aged 12-17 years who had received complete primary schedule of an inactivated vaccine (last dose applied 6-18 months before booster), the incidence rate of AEs until 28 days after booster was 82.7%; with local pain as the most frequent AEs reported (57.3%), followed by myalgia (40.0%) [15]. In a phase III study with another subunit vaccine, Nuvaxovid, administration of a booster dose approximately nine months after the primary series in adolescents aged 12-17 years resulted in injection site tenderness, headache, fatigue, injection site pain, muscle pain and malaise with reported frequencies ranging from 47% to 72% within seven days after the booster [16]. It should be noted that comparisons of AEs rates across are limited by differences in their adjuvant and antigen composition. Nuvaxovid employs Matrix-M™, and IndoVac® uses a combination of alumina and CpG 1018—adjuvants potentially more reactogenic than the alumina-only formulation used in SOBERANA® Plus. Despite these differences, the data consistently support the established safety of the protein subunit vaccine platform.”

Comments 4: Terminology (Line 43, Abstract/Conclusions) The manuscript assessed only short-tern immunogenicity at a single post-booster timepoint, without long-term follow-up or efficacy/effectiveness data. The phrases such as “sustain broad immune protection” may therefore be overstated. Consider rephrasing to terms such as “stimulate” or “maintain broad immunity,” or similar wording more directly aligned with your findings

Response 4: Thank you for pointing this out, you are right. We have rephrased “sustain broad immune protection” to “maintain broad immunity,” as you suggested. (see page 1, line 44 in the corrected version)

3. Errors and format.

3.1.  Subscripts: Please use consistent formatting with subscripts throughout (e.g., cVNT₅₀, mVNT₅₀, TCID₅₀, IC₅₀), including figure labels

Response 3.1: Thank you for pointing this out. We have corrected that issue in the whole manuscript (text, supplementary file, tables and figures)

3.2.  Units: Standardise units according to SI conventions. For example, “L” should be used consistently for litre, avoiding variation between “l” and “L” (e.g., μL vs μl; mol/L vs mol/l).

Response 3.2: Thank you for pointing this out. We have corrected that issue in the whole manuscript (text, supplementary file, tables and figures)

3.3.  Figures 2C and 2D: Both panels present cVNT₅₀ data but against different virus strains. Standardising the y-axis scale would allow a clearer visual comparison across panels.

Response 3.3:. Both y-axes are standardized allowing visual comparison across panels.  

3.4.  Capitalization: Ensure consistent use of “Omicron” (capitalized as a proper noun) in all figures, tables, and text. For instance, corrections are needed at lines 117, 122, 123, 230, 239, 246, 283, and elsewhere.

Response 3.4: Thank you for pointing this out. We have corrected that issue in the whole manuscript (text, supplementary file, tables and figures)

3.5.  Decimal separator: Please use the period (“.”) as the decimal separator instead of the comma (“,”) to align with the journal’s formatting style.

Response 3.5: Thank you for pointing this out. We have corrected that issue in the whole manuscript (text, supplementary file, tables and figures)

3.6.  Table S3 (Supplementary material): The three subgroup columns are all labelled “3–11 y/o.” This appears to be an error. They should likely read 3–11 y/o, 12–18 y/o, and Total. Please verify and correct.

Response 3.6: Thank you for pointing this out. It was a mistake. We have corrected it in the current version.
